# Recent Advances of ^68^Ga-Labeled PET Radiotracers with Nitroimidazole in the Diagnosis of Hypoxia Tumors

**DOI:** 10.3390/ijms241310552

**Published:** 2023-06-23

**Authors:** Anh Thu Nguyen, Hee-Kwon Kim

**Affiliations:** 1Department of Nuclear Medicine, Jeonbuk National University Medical School and Hospital, Jeonju 54907, Republic of Korea; thu.ngnanh39@gmail.com; 2Research Institute of Clinical Medicine of Jeonbuk National University-Biomedical Research Institute of Jeonbuk National University Hospital, Jeonju 54907, Republic of Korea

**Keywords:** Ga-68, hypoxia, PET, radiopharmaceuticals, tumor

## Abstract

Positron emission tomography (PET) is a noninvasive molecular imaging method extensively applied in the detection and treatment of various diseases. Hypoxia is a common phenomenon found in most solid tumors. Nitroimidazole is a group of bioreducible pharmacophores that selectively accumulate in hypoxic regions of the body. Over the past few decades, many scientists have reported the use of radiopharmaceuticals containing nitroimidazole for the detection of hypoxic tumors. Gallium-68, a positron-emitting radioisotope, has a favorable half-life time of 68 min and can be conveniently produced by ^68^Ge/^68^Ga generators. Recently, there has been significant progress in the preparation of novel ^68^Ga-labeled complexes bearing nitroimidazole moieties for the diagnosis of hypoxia. This review provides a comprehensive overview of the current status of developing ^68^Ga-labeled radiopharmaceuticals with nitroimidazole moieties, their pharmacokinetics, and in vitro and in vivo studies, as well as PET imaging studies for hypoxic tumors.

## 1. Introduction

Molecular imaging refers to the utilization of specialized imaging for biological processes at the molecular or cellular level in living subjects, enabling the comprehensive understanding of complex biological phenomena [1]. Molecular imaging technologies have emerged as powerful tools that offer valuable insights into biological events, disease pathology, and underlying mechanisms. As a result, they have become increasingly indispensable in diverse fields such as neuroscience, drug therapy assessment, oncology, and numerous others [2,3,4,5,6,7,8,9,10,11,12,13,14]. 

To enable visualization of biological events, a range of imaging techniques has been developed. These encompass anatomic imaging methods (magnetic resonance imaging (MRI) and computed X-ray tomography (CT), optical imaging methods, and nuclear imaging methods (single photon emission computed tomography (SPECT) and positron emission tomography (PET)), as well as hybrid imaging methods (PET/CT, SPECT/CT, and PET/MR) which combine the strengths of different modalities to provide enhanced imaging capabilities [15,16,17,18,19].

PET, a noninvasive imaging technique using position-emitting radioisotopes, visualizes, characterizes, and quantifies physiological and biological processes [20]. PET has found extensive utility in both biological and clinical applications since its initial description in 1950 [21]. This is primarily due to its remarkable sensitivity and ability to penetrate deeply into tissues [4]. It has proven to be effective in numerous studies of various diseases such as inflammatory diseases, heart diseases, Alzheimer’s diseases, and many types of cancer [22,23,24,25,26,27,28,29,30,31,32,33,34,35,36,37,38,39,40].

In oncology, PET imaging methods detect tumors as small as one millimeter, providing a significant advantage over conventional anatomic imaging techniques such as MRI and CT [41,42]. This high sensitivity allows for the early detection and precise localization of tumors, facilitating more effective diagnosis and treatment planning. The utilization of PET in the detection of cancer significantly enhances cancer diagnosis and staging, while also enabling the monitoring of drug responses [43,44]. By providing valuable information on tumor metabolism and response to treatment, PET contributes to improved patient care and outcomes in the field of oncology. PET imaging studies heavily rely on the design and preparation of suitable imaging probes, commonly referred to as radiopharmaceuticals. These radiopharmaceuticals contain positron-emitting radionuclides [45,46,47,48]. 

Hypoxia is a condition characterized by insufficient oxygen (O_2_) supply to tissues, impeding normal biological functions [49,50]. In cancer, the continuous growth of tumor cells results in abnormalities in tumor blood vessels, thus reducing the oxygen supply in the tumor interior [50,51]. Hypoxia is a prevalent characteristic observed in the majority of solid tumors, often resulting in oxygen levels below 1000 parts per million (ppm) [52]. Extensive research has shown that hypoxia plays a significant role in the malignant progression and metastases of various cancers [53,54,55,56,57,58,59,60,61], resistance against therapies, and cancer prognosis [62,63,64,65,66,67]. The detection of tumor hypoxia can be achieved through invasive methods, such as the direct insertion of an oxygen microelectrode into the tumor to measure oxygen tension (pO_2_) [68], the expression of hypoxia-inducible factor (HIF) on biopsy specimens [69,70,71], or the comet assay [72,73,74]. Nevertheless, these invasive methods have several drawbacks, including the need for small tumor samples and easily accessible tumors, specialized personnel, and equipment. Consequently, there has been a growing emphasis on the development of noninvasive methods that utilize radiolabeled imaging agents to overcome these limitations [75], allowing for a more comprehensive assessment of tumor hypoxia in a broader range of patients.

Nitroimidazole has emerged as a significant structure for hypoxia detection [76,77]. The moiety is known to be bioreducible. Upon diffusion into viable cells, nitroimidazoles undergo selective reduction, forming reactive nitro radical anions. This process, known as the activation reaction, facilitates their intracellular uptake and retention. The activation reaction is reversible under normal oxygen conditions, and the nitro radical anions generated can undergo reoxidation to form nonradical compounds. However, in hypoxic conditions, additional reductions of nitro radical anions occur, generating reactive species that irreversibly bind to cellular macromolecules [76,78]. This distinction allows for the differentiation of hypoxic tissues from normoxic tissues. 

Nitroimidazole-containing compounds have been utilized for the detection of hypoxia, specifically in PET imaging studies. As a result, significant efforts have been devoted to the development of radiolabeled compounds with nitroimidazole moieties. [^18^F]FMISO is the most common PET tracer for hypoxia imaging owing to its simple synthetic process and selectivity toward hypoxia. [^18^F]FMISO has been widely studied for hypoxia imaging of many types of cancers in both preclinical and clinical studies [79,80,81,82,83,84]. In the development of novel PET imaging agents, [^18^F]FMISO is still the standard for hypoxia and used as the control for hypoxia in many studies [85,86,87]. However, [^18^F]FMISO has some drawbacks such as slow blood clearance and high uptake values in the liver and gut due to its lipophilicity (log*p* value of about 0.4) and slow tumor accumulation [50]. These drawbacks result in low-contrast PET images and extended optimal acquisition time, limiting its utility in clinical studies [88,89]. Thus, in order to improve the sensitivity and convenience of [^18^F]FMISO, the second generation of PET imaging agents for hypoxia was developed, namely [^18^F]FAZA and [^18^F]FETNIM. These radiotracers are also analogues of 2-nitroimidazole but are more hydrophilic compared to [^18^F]FMISO [90,91]. [^18^F]FAZA, in particular, exhibited good uptakes and acceptable tumor/blood ratios in patients with lung, head, and neck tumors, lymphoma, and gliomas [92]. Along with [^18^F]FAZA, [^124^I]IAZA is also another alternative to [^18^F]FMISO with better tumor/blood contrasts at an early time post-injection [93,94,95,96]. [^18^F]FETNIM was also tested clinically in patients with lung cancer, glioma, cervical carcinoma, esophageal, and pancreatic cancer [97]. For example, the use of [^18^F]FETNIM PET/CT has enabled the visualization of hypoxia in patients with esophageal squamous cell carcinoma. Recently, when tested in patients with head and neck cancer, [^18^F]FETNIM PET/CT was also successfully used for the selection of patients with hypoxia predicting poor overall survival [98]. However, when compared to [^18^F]FMISO in clinical studies in patients with lung cancer, [^18^F]FETNIM failed to show higher tumor/blood ratios than [^18^F]FMISO [99] (Table 1). 

Additionally, [^18^F]EF5 was developed as a fluorinated radiotracer more lipophilic than [^18^F]FMISO and the other second-generation radiotracers, with a log*p* value of 5.7 which enables faster distribution into normal and tumor tissues yet slower clearance from blood [100]. [^18^F]EF5 is a potential PET tracer with favorable tumor/muscle ratios when used for the imaging of hypoxia in patients with head and neck cancers. [^18^F]EF3, an analogue of [^18^F]EF5 with medium lipophilicity (log*p* = 1.25) was also reported. However, when tested in mice and rat models, [^18^F]EF3 was not superior to [^18^F]FMISO in tumor/muscle ratios [101,102,103]. 

Later, [^18^F]HX4 and [^18^F]FRP170 were prepared as PET tracers for hypoxia. [^18^F]HX4 was clinically tested in patients with lung, head and neck, esophageal, and pancreatic cancer, and showed a comparable effective dose, improved contrast, and more favorable hydrophilicity (log*p* = −0.69) in comparison to [^18^F]FMISO; thus, frequent bladder voiding was required for patients due to the rapid renal clearance characteristic of [^18^F]HX4 [104,105]. [^18^F]FRP170, which contains hydrophilic groups, was more hydrophilic than [^18^F]FMISO (log*p* = 0.094). It demonstrated good tumor uptake and tumor/background contrast in PET images as well as rapid clearance via the kidneys when tested in humans with lung cancers [106,107]. 

Moreover, Cu-ATSM, a lipophilic hypoxia radiotracer not containing nitroimidazole moieties, was developed. It can be radiolabeled with several radionuclides like ^64^Cu, ^60^Cu, and ^62^Cu. When tested in patients with cervical carcinoma, ^64^Cu-ATSM showed better-quality PET images with lower noise compared to ^60^Cu-ATSM [108]. Furthermore, numerous novel radiotracers have been developed to enhance the efficacy of current radiopharmaceuticals, and numerous preclinical studies have been conducted. 

Generally, ^18^F-labeled radiotracers have been extensively studied due to their appropriate half-life time of 109.8 min and the ability to be incorporated into small and large molecules [109]. Moreover, ^68^Ga is produced via the use of ^68^Ge/^68^Ga generators. Therefore, the preparation of ^68^Ga-labeled radiopharmaceuticals has become more convenient compared to other positron emitters like fluorine-18, carbon-11, or copper-64 which require a cyclotron. For example, despite the widespread use of ^18^F-labeled radiotracers in PET imaging, radiosyntheses of ^18^F-labeled radiotracers mostly rely on the production of fluorine-18 using cyclotron, which is large in size and not widely available [110]. 

Developing more hydrophilic radiotracers compared to [^18^F]FMISO is currently the primary need in the development of novel hypoxia-targeting radiopharmaceuticals. ^68^Ga, a radionuclide with a half-life of 68 min, has also been applied for PET imaging. Bifunctional chelates have been widely employed to generate octahedral coordination with the gallium-68 radionuclide at the same time to bear the nitroimidazole moieties. These chelates often contain hydrophilic moieties which contribute to increasing the hydrophilicity of the ^68^Ga-labeled complex. Recently, ^68^Ga has gained significant attention, primarily because ^68^Ga can be easily produced using bench-top Germanium-68/Gallium-68 (^68^Ge/^68^Ga) generators. The preparation and formulation of ^68^Ga-labeled radiopharmaceuticals can be conveniently carried out on-site using a commercial kit [111,112]. The accessibility and convenience have contributed to the growing interest in ^68^Ga as a promising option for PET imaging (Figure 1). 

This review provides a comprehensive summary of the developments made in ^68^Ga-labeled PET studies focused on tumor hypoxia since 2009. The review also covers various aspects, including the physicochemical properties of these radiopharmaceuticals, in vitro biological evaluations, in vivo biodistribution studies, and PET imaging results. By presenting a comprehensive overview, this review aims to provide insights into the progress and potential of ^68^Ga-labeled PET radiotracers in the field of tumor hypoxia research.

## 2. Results

In 2009, Mukai and co-workers developed a bifunctional radiopharmaceutical ^67^Ga-DOTA-MN2 ([^67^Ga]**3**), based on the utilization of DOTA chelate (2,2′,2″,2′″-(1,4,7,10-Tetraazacyclododecane-1,4,7,10-tetrayl)tetraacetic acid) for coordination with gallium and metronidazole groups as hypoxia targeting moieties [113]. The precursor DOTA-MN2 (**2**) could be synthesized from cyclen (**1**) via a seven-step procedure (Figure 1). The radiolabeling reaction between **2** and ^67^Ga-citrate was conducted in a buffer of ammonium acetate at 95 °C and produced radiolabeled complex [^67^Ga]**3** with radiochemical yields ranging from 35% to 59% and radiochemical purity exceeding 96%.

[^67^Ga]**3** remained intact in vitro in saline and mouse plasma for 24 h. An in vivo biodistribution study conducted in normal mice showed rapid blood clearance of [^67^Ga]**3** with 4% in the kidney and <0.9% in other tissues at 30 min post-injection (p.i.). Compared to the commonly used radiopharmaceutical ^67^Ga-citrate, [^67^Ga]**3** exhibited lower nonspecific retention in normal mice. In C3H/He mice bearing NFSa tumors, [^67^Ga]**3** demonstrated remarkable accumulation in tumor tissues (tumor uptake = 0.20 ± 0.04 %ID/g) compared to normal tissues (except for the kidneys) such as the stomach, liver, muscle, intestine, spleen, and blood (ranging from 0.05 ± 0.01 to 0.11 ± 0.03 %ID/g). When compared to ^67^Ga-DOTA without hypoxia-targeting moiety (tumor uptake = 0.23 ± 0.04 %ID/g at 1 h p.i.), [^67^Ga]**3** exhibited higher tumor uptake at 1 h p.i. (0.49 ± 0.12 %ID/g), indicating the contribution of metronidazole groups to targeting hypoxic regions. Due to its improved tumor uptake, rapid blood clearance, and specific tumor accumulation, [^67^Ga]**3** exhibited good tumor/blood and tumor/muscle ratios (4.55 ± 0.44 and 4.42 ± 1.45 at 6 h p.i.).

In 2010, Jeong and co-workers synthesized two ^68^Ga-labeled complexes ^68^Ga-NOTA-NI ([^68^Ga]**7**) and ^68^Ga-SCN-NOTA-NI ([^68^Ga]**8**) containing 2-nitroimidazole binding to the bifunctional chelates NOTA and SCN-NOTA [114]. From 2-nitroimidazole, precursors NOTA-NI and SCN-NOTA-NI were prepared in three steps (Figure 2). The radiolabeling reactions in boiling water at pH 3 with ^68^GaCl_3_ (obtained from ^68^Ge/^68^Ga generator) generated the complexes [^68^Ga]**7** and [^68^Ga]**8** in 96–96.3% radiolabeling yields. Moreover, [^68^Ga]**8** with three free carboxyl groups was believed to form an octahedral structure with gallium, while the structure of [^68^Ga]**7** (with two free carboxyl groups) in which the oxygen of amide group coordinated with gallium was confirmed by crystallography.

[^68^Ga]**7** and [^68^Ga]**8** were both hydrophilic in which [^68^Ga]**8** had a higher log*p* value than [^68^Ga]**7** (−2.27 for [^68^Ga]**8** and −2.71 for [^68^Ga]**7**, respectively) due to the presence of the benzene ring in the structure. Both ^68^Ga-labeled complexes remained intact in prepared solutions and in human serum, and exhibited favorably low protein binding. In vitro cellular uptake studies of [^67^Ga]**7** and [^68^Ga]**8** in ovarian cancer cell line CHO and colon cancer cell line CT-26 showed higher uptake values under hypoxic environments than normoxic environments at 1 h. In an in vivo study using CT-26 xenografts-bearing mice, [^68^Ga]**7** and [^68^Ga]**8** were found to be excreted mostly via the renal route. In tumors, both complexes exhibited the highest uptakes at 10 min p.i. (2.47 ± 0.47 %ID/g for [^68^Ga]**7** and 2.37 ± 0.29% ID/g for [^68^Ga]**8**). At 1 h p.i., the tumor uptakes of [^68^Ga]**7** and [^68^Ga]**8** were reduced to 0.73 ± 0.18 and 0.61 ± 0.06 %ID/g, respectively. When compared to the common ^18^F-labeled radiotracers containing 2-nitroimidazole [^18^F]FAZA and [^18^F]FMISO, [^68^Ga]**7** and [^68^Ga]**8** had lower tumor/blood ratios yet similar tumor/muscle ratios. In the PET imaging study, at 1 h p.i., accumulation of [^68^Ga]**7** showed a better contrast with the standardized uptake value (SUV) of 5.7 ± 2.5 than [^68^Ga]**8** (3.95 ± 1.3). Moreover, both complexes showed higher SUV than [^18^F]FAZA and [^18^F]FMISO. 

In 2011, Jeong and co-workers developed another two ^68^Ga-labeled derivatives of 2-nitroimidazole in which the 2-nitroimidazole moieties were connected to the bifunctional chelate DOTA also via an amide bond ([^68^Ga]**12**) or a thiourea bond ([^68^Ga]**13**) [115]. The precursors **10** and **11** were prepared in three steps from 2-nitroimidazole. These precursors were successfully radiolabeled with ^68^Ga^3+^ to produce radiotracers [^68^Ga]**12** and [^68^Ga]**13** with radiochemical yields over 98% (Figure 3). The specific activity of [^68^Ga]**12** was 4.81 × 10^6^ GBq/mol, and that of [^68^Ga]**13** was 7.77 × 10^6^ GBq/mol. Partition coefficients of [^68^Ga]**12** (−4.6) and [^68^Ga]**13** (−4.5) were below zero, indicating hydrophilicity. The stabilities of [^68^Ga]**12** and [^68^Ga]**13** in prepared solutions and human serum for 2 h were demonstrated, along with desirable protein bindings.

In vitro cellular uptake studies conducted in Hela, CHO, and CT-26 cancer cell lines revealed higher uptake values under hypoxic than normoxic environments for both complexes [^68^Ga]**12** and [^68^Ga]**13**, in which uptake values of the complexes in CT-26 cell lines were the highest. In CT-26 xenografts-bearing mice, both [^68^Ga]**12** and [^68^Ga]**13** were excreted via the kidneys, obtaining the highest uptakes. [^68^Ga]**12** and [^68^Ga]**13** exhibited the highest tumor uptakes at 10 min p.i. (3.17 %ID/g and 2.78 %ID/g, respectively) and decreased to 0.64 %ID/g and 0.59 %ID/g, respectively, at 2 h p.i. Furthermore, due to rapid clearance from blood and muscle, tumor/blood and tumor/muscle ratios of [^68^Ga]**12** and [^68^Ga]**13** were relatively high at 2 h p.i. A PET imaging study in mice bearing CT-26 xenografts indicated a significantly higher tumor uptake of [^68^Ga]**12** (SUV = 0.53 ± 0.1 at 1 h p.i.) compared to [^68^Ga]**13** (SUV = 0.17 ± 0.1 at 1 h p.i.). In addition, [^68^Ga]**12** also showed better contrast with a tumor/nontumor ratio of 5.64 ± 0.8 at 1 h p.i., higher than that of [^68^Ga]**13** (3.83 ± 0.8 at 1 h p.i.). 

In 2013, Rey and co-workers synthesized two 5-nitroimidazole derivatives **15** (10-[2-(2-methyl-5-nitro-1H-imidazole-1-yl)ethylaminocarbonylmethyl]-1,4,7,10-tetraazacyclododecane-1,4,7-triacetic acid) and **16** (10-{[N-methyl-1-[1-(2-(2-methyl-5-nitro-1 H-imidazole-1-yl)ethyl)-1H-1,2,3-triazole-4yl]methylaminocarbonylmethyl}-1,4,7,10-tetraazacyclododecane-1,4,7-triacetic acid), conjugating with DOTA chelates in order to form complexes with ^68^Ga ([^68^Ga]**17** and [^68^Ga]**18**) [116]. From mono-NHS-tris-t-butyl-DOTA ester (**14**), ligands **15** and **16** were synthesized in two steps (Figure 4). The obtained ligands were then employed in the radiolabeling reactions with GaCl_3_ in sodium acetate solution at 95 °C to generate ^68^Ga complexes [^68^Ga]**17** and [^68^Ga]**18** with specific activities of 67± 23 MBq/nmol for both complexes. Good radiochemical purities (>90%) of the two ^68^Ga complexes were demonstrated by using HPLC.

[^68^Ga]**17** and [^68^Ga]**18** were stable in labeling milieu and human plasma and showed low protein binding. Moreover, when incubated with chelating agent DTPA, two ^68^Ga complexes remained intact and no trans-chelation was detected. [^68^Ga]**17** and [^68^Ga]**18** were found to be hydrophilic (log*p* = −1.65 ± 0.05 and −3.30 ± 0.10, respectively) and both complexes exhibited higher hydrophilicity than [^18^F]FMISO (log*p* = −0.40 ± 0.03), enabling faster clearance from normal organs and tissues. In in vitro experiments using HCT-15 cell lines, [^68^Ga]**17** and [^68^Ga]**18** were selectively uptaken in hypoxic conditions. Notably, [^68^Ga]**17** exhibited a higher hypoxic/normoxic ratio than both [^18^F]FMISO and [^68^Ga]**18**. Biodistribution of the two complexes in induced Lewis carcinoma-bearing C57 mice suggested rapid blood and liver clearance, as well as renal excretion owing to their high hydrophilicity. [^68^Ga]**17** and [^68^Ga]**18** showed similar initial tumor uptake values (1.31 ± 0.93 and 1.34 ± 0.55 at 0.5 h p.i., respectively). However, [^68^Ga]**18** exhibited good retention in tumors, while only 50% radioactivity of [^68^Ga]**17** was retained at 2 h p.i. Compared to [^18^F]FMISO (T/M = 4.4 ± 1.0 at 2 h p.i.), both [^68^Ga]**17** and [^68^Ga]**18** exhibited significantly higher tumor/muscle ratios at 2 h p.i. (5.1 ± 1.7 and 6.6 ± 1.6, respectively) due to their fast clearance from soft tissues. 

In 2013, Mukai and co-workers continued to study ^67/68^Ga-DOTA-MN2 complexes ([^67/68^Ga]**22**) containing metronidazole and DOTA chelator [117]. They also synthesized the complex ^67^Ga-DOTA-MN1 ([^67^Ga]**23**) bearing one metronidazole moiety. The ligands DOTA-MN1(**21**) and DOTA-MN2 (**20**) were synthesized from DOTA-tris(t-Bu) ester via two-step processes. The ligands were then radiolabelled with ^67^GaCl_3_ or ^68^GaCl_3_ to produce the complexes [^67/68^Ga]**22** and [^67^Ga]**23** in high-to-excellent radiochemical yields and >99% radiochemical purities (Figure 5). Moreover, ^67^Ga-DOTA and ^68^Ga-DOTA were also prepared in >99% radiochemical yields and excellent radiochemical purities.

The initial tumor uptakes of [^67^Ga]**22** and [^67^Ga]**23** in C3H/He mice bearing FM3A tumors were similar at 1 h p.i. (0.52 %ID/g and 0.50 %ID/g, respectively). However, due to the fast blood clearance of [^67^Ga]**22**, this complex exhibited higher tumor/blood and tumor/muscle ratios than [^67^Ga]**23**. The results of [^67^Ga]**22** in ex vivo autoradiography and immunohistochemistry for pimonidazole in mice bearing FM3A tumors were compared with ^67^Ga-DOTA. Complex ^67^Ga-DOTA did not show consistency between autoradiography results and pimonidazole-positive regions. In contrast, [^67^Ga]**22** complex accumulated in positive regions in immunostaining for pimonidazole, indicating accumulation in hypoxic regions. PET imaging of [^68^Ga]**22** and ^68^Ga-DOTA in mice bearing FM3A tumors also suggested a clearly observed tumor in the mice injected with [^68^Ga]**22** at 1 h p.i., while clear uptake of ^68^Ga-DOTA in tumor sites was not detected. 

In 2015, Jeong and co-workers developed four ^68^Ga-labeled trivalent complexes ([^68^Ga]**28**–**31**) containing one or several nitroimidazole moieties for hypoxia targeting, and 1,4,7-triazacyclononane-1,4,7-tris[methyl(2-carboxyethyl)phosphinic acid (TRAP) was used as a bifunctional chelating agent [118]. TRAP ligand was synthesized using the method reported by Sun and co-workers [119]. Coupling reactions between TRAP and 2-(2-nitroimidazolyl)ethylamine generated precursors **25**, **26**, and **27** bearing one, two, and three nitroimidazole moieties, respectively (Figure 6). TRAP ligand and precursors **25**–**27** were then radiolabeled with ^68^Ga^3+^ in sodium acetate buffer (pH 4.5) at 95 °C to afford ^68^Ga complexes [^68^Ga]**28**, [^68^Ga]**29**, [^68^Ga]**30**, and [^68^Ga]**31** with radiochemical yields >96%, radiochemical purities >99%, and specific activities of 0.229 ± 0.103, 0.229 ± 0.103, 0.262 ± 0.050, and 0.291 ± 0.071 MCi/mol, respectively. 

Complexes [^68^Ga]**30** and [^68^Ga]**31** were found to be hydrophilic with log*p* values of −3.64 and −3.28. Although the log*p* values of [^68^Ga]**28** and [^68^Ga]**29** were undetectable, they were indicated to be hydrophilic due to the presence of free carboxylic groups in the structures. Four ^68^Ga-labeled complexes [^68^Ga]**28**–**31** remained stable in vitro and exhibited lower protein bindings than ^68^Ga-NOTA-SCN-NI, ^68^Ga-DOTA-NI, and ^68^Ga-DOTA-SCN-NI bearing one nitroimidazole moiety. When tested for cellular uptake in U87MG and CT-26 cell lines, among the four complexes, [^68^Ga]**31** showed increasing cellular uptakes in hypoxic conditions and the highest hypoxic/normoxic ratios at 15, 30, and 60 min p.i. Biodistribution studies of four complexes in BALB/c mice bearing CT-26 xenograft suggested that complexes containing nitroimidazole [^68^Ga]**29**–**31** were selectively uptaken in tumor regions, in which [^68^Ga]**31** retained the most in tumor cells at 60 min p.i. Moreover, [^68^Ga]**31** also had the highest tumor/muscle and tumor/blood ratios. In PET images of mice bearing CT-26 xenograft, [^68^Ga]**31** indicated the best contrast among the four complexes [^68^Ga]**28**–**31** and even the reported complexes containing one nitroimidazole with an SUV value of 0.59 ± 0.09. 

The radiotracer ^68^Ga-**31** was evaluated on 20 women with cervical cancer by using PET/CT imaging and immunohistochemistry and compared with ^18^F-FDG for selectivity for tumor hypoxia by Sathekge and co-workers in 2022 [120]. The patients were injected with ^68^Ga-**31**, with injected activity ranging from 1.8 to 2.2 MBq/kg. Pelvic images were taken at 30 min p.i. and the PET/CT images were taken at 60 min p.i. after urinary catheterization. ^18^F-FDG PET/CT studies were performed on the same 20 patients on separate days to ^68^Ga-nitroimidazole PET/CT, with the injected activity of 3–5 MBq/kg. Biodistribution studies demonstrated that, of all 20 patients, 12 patients showed uptake values of ^68^Ga-nitroimidazole higher than background uptakes, indicating hypoxia. However, among 15 patients in the immunohistochemical study, only 5 patients exhibited strong HIF-1α positive regions. Thus, from PET imaging results, tumor/muscle ratios of ^68^Ga-nitroimidazole and ^18^F-FDG exhibited no significant correlation with the immunohistochemical HIF-1α positive regions, which might be explained by the heterogeneity of hypoxia, resulting in a smaller volume of hypoxia than the tumor volume. Notably, hypoxic tumor volume was negatively correlated with immunohistochemistry, which might be due to the feedback mechanism that occurred between HIF-1α and markers or genes of the tumor, or the presence of the other isoforms of HIF-1α, such as HIF-2α and HIF-3α. 

In 2015, Orvig and co-workers reported nine ^68^Ga-labeled complexes containing nitroimidazole moieties and ligand H_2_dedpa or its cyclohexyl derivative H_2_*CHX*dedpa [121]. 1-(ω-Bromoalkyl)nitroimidazoles prepared from the corresponding nitroimidazole were conjugated with H_2_dedpa or H_2_*CHX*dedpa to produce the desired ligands **38**–**43** and **44**–**46** in two steps (Figure 7). H_2_dedpa-N,N′-alkyl-NI and H_2_*CHX*dedpa-N,N′-propyl-NI ligands were radiolabeled with ^67^Ga and ^68^Ga in >99% radiochemical yields and good radiochemical purities in an HPLC radiochromatogram when using ligand concentrations of 10^−5^ and 10^−4^ M.

When nine ^67^Ga complexes [^67^Ga]**47**–**55** were incubated with apo-transferrin, which acted as a competitor for gallium (III), they remained intact with good stability (86–90%) and exhibited favorably low protein binding and with no *trans*-chelation observed. In vitro cellular uptakes of radiotracers [^68^Ga]**50**–**52** and [^68^Ga]**55** in three cancer cell lines HT-29, LCC6^HER-2^, and CHO indicated higher uptakes in hypoxic environments than those in normoxic environments for all tested cell lines. Furthermore, in comparison to ^68^Ga-DOTA and ^68^Ga-DOTA-NI tracers (hypoxic/normoxic ratios ranging from 1.5 to 5.6 at 1 h) [87,88], ^68^Ga complexes with *CHX*dedpa showed higher hypoxic/normoxic ratios (4.7–7.3 at 1 h). Moreover, the position of the nitro group of nitroimidazole did not affect the cellular uptake and retention of ^68^Ga-(*CHX*)dedpa-NI.

In 2015, Sherry and co-workers synthesized a ^68^Ga-labeled complex bearing 2-nitroimidazole and HP-DO3A ligand (^68^Ga-HP-DO3A-NI, [^68^Ga]**58**) [122]. Ligand HP-DO3A-NI (**57**) could be synthesized in two steps from DO3A-tris-*t-*butyl ester (**56**) and (S)-oxiranylmethylnitroimidazole. [^68^Ga]**58** was prepared in 85% radiochemical yield via a radiolabeling reaction between the ligand **57** and ^68^Ga^3+^ conducted in sodium acetate buffer (pH 3.5) at 99 °C (Figure 8). [^68^Ga]**58** had good radiochemical purity (>95%) and a specific activity of 1.2 × 107 GBq/mol.

[^68^Ga]**58** had a log*p* value of −4.6 ± 0.1, indicating hydrophilicity. An in vitro cellular uptake study of [^68^Ga]**58** in A549 cancer cell lines showed significantly higher accumulation under N_2_ atmosphere (hypoxia) than under air atmosphere with hypoxia/normoxia ratios of 3.7–9.3 from 30 min to 120 min, which were comparable to those of [^68^Ga]-DOTA-NI in CT-26 cancer cell lines. After hypoxia in A549 tumor-bearing SCID mice was confirmed, the in vivo biodistribution of [^68^Ga]**58** and PET imaging studies were evaluated. [^68^Ga]**58** was selectively uptaken at the maximum in the tumor at 10 min p.i. and showed good retention until 2 h p.i., while other organs and tissues except the kidneys had low uptake values. Due to the rapid clearance from muscle, [^68^Ga]**58** exhibited a high tumor/muscle ratio of 5.0 ± 1.2 at 2 h p.i. The control radiotracer ^68^Ga-HP-DO3A with no nitroimidazole moiety, in contrast, showed no accumulation in the tumor. Moreover, applying 100% O_2_ as the carrier gas in anesthesia of tumor-bearing mice led to a decrease in tumor uptake of [^68^Ga]**58** in PET imaging compared to using air, indicating the consistency between hypoxia and tumor uptake of [^68^Ga]**58**. 

In 2021, Shimizu and co-workers reported the synthesis of four ^68^Ga-labeled complexes [^68^Ga]**67,** [^68^Ga]**68,** [^68^Ga]**70,** and [^68^Ga]**71** ([^68^Ga]DN-3, [^68^Ga]DN-4, [^68^Ga]NN-3, and [^68^Ga]NN-4, respectively) containing 2-nitroimidazole and a bifunctional chelate (DOTA or NOTA), which were connected via a linker [123]. The four ^68^Ga-labeled complexes were compared with the previously reported complexes [^68^Ga]DN-2 ([^68^Ga]**66** containing DOTA) and [^68^Ga]NN-2 ([^68^Ga]**69** containing NOTA) to study the effect of the length of the linkers. Precursors **60**–**62** and **63**–**65** were synthesized from 2-nitroimidazole, N-Boc-alkyl bromides, and bifunctional chelator *p*-SCN-Bn-DOTA or *p*-SCN-Bn-NOTA in two steps. These were then radiolabeled with ^68^Ga via the coordination of bifunctional chelators and ^68^GaCl_3_ in a solution of acetate buffer and ascorbic acid to produce the desired ^68^Ga-labeled complexes with 27.2–63.8% RCYs, 128–153 GBq/μg specific activities, and >95% radiochemical purities (Figure 9).

Four complexes [^68^Ga]**67**, [^68^Ga]**68**, [^68^Ga]**70**, and [^68^Ga]**71** showed selectivity for hypoxia in vitro in FaDu cancer cell lines; in particular, cellular uptakes under hypoxic conditions were significantly higher than those under normoxic conditions. Compared to the previously reported complexes [^68^Ga]**66** and [^68^Ga]**69**, the cellular uptakes of the four ^68^Ga complexes were significantly improved under hypoxic conditions, indicating the effects of linker length on connecting nitroimidazole and bifunctional chelate on in vitro hypoxia selectivity. In vivo biodistribution studies of the four complexes in Balb/c mice bearing FaDu xenograft showed that, at 2 h p.i., [^68^Ga]**67** exhibited the highest accumulation in tumors (0.56 ± 0.18 %ID/g) among the four complexes (0.11 ± 0.04 %ID/g for [^68^Ga]**68**, 0.10 ± 0.04 %ID/g for [^68^Ga]**70**, and 0.15 ± 0.11 %ID/g for [^68^Ga]**70**). In addition, [^68^Ga]**67** also exhibited higher tumor uptake at 2 h p.i. than the previously reported complexes [^68^Ga]**66** (0.27 ± 0.02 %ID/g) and [^68^Ga]**69** (0.27 ± 0.10 %ID/g). Tumor/blood and tumor/muscle ratios of the four ^68^Ga complexes were not significantly different from [^68^Ga]**66** and [^68^Ga]**69** with tumor/blood ratios over one and tumor/muscle ratios over three, which were acceptable. Biodistribution studies also showed a high accumulation of the four complexes in the intestine. In the PET/CT image studies of [^68^Ga]**67**, [^68^Ga]**68**, [^68^Ga]**70**, and [^68^Ga]**71**, tumors were observed. High accumulation of the four ^68^Ga complexes was also observed in the intestinal region, which was consistent with biodistribution studies.

In 2021, Mallia and co-workers synthesized ^68^Ga complexes ^68^Ga-DOTAGA-2-NIM ([^68^Ga]**75**) and ^68^Ga-NODAGA-2-NIM ([^68^Ga]**76**) bearing bifunctional chelators DOTAGA and NODAGA [124]. Precursors DOTAGA-2-NIM (**73**) and NODAGA-2-NIM (**74**) could be prepared via a three-step procedure from 2-nitroimidazole. [^68^Ga]GaCl_3_ was added into a sodium acetate buffer solution (pH 4, 100 °C) containing the precursors to produce the ^68^Ga-labeled complexes [^68^Ga]**75** and [^68^Ga]**76** (Figure 10).

Both complexes [^68^Ga]**75** and [^68^Ga]**76** exhibited hydrophilicity with nearly similar log*p* values of −2.42 ± 0.19 and −2.62 ± 0.14, respectively. Although both ^68^Ga complexes were stable in human serum, [^68^Ga]**76** had a stronger affinity for serum proteins than [^68^Ga]**75**, which was explained by the large cavity size of the DOTA ligand, allowing for the coordination from another donor group of serum proteins to the gallium ion. In vitro cellular uptake of [^68^Ga]**75** and [^68^Ga]**76** in CHO cell lines showed selectivity for hypoxia with hypoxia/normoxia ratios over one. Particularly, [^68^Ga]**76** had a hypoxia/normoxia ratio of 2.88 ± 0.36 at 3 h, higher than that of [^68^Ga]**75**. In fibrosarcoma-tumor-bearing Swiss mice, [^68^Ga]**75** and [^68^Ga]**76** exhibited good initial uptake values in tumors (5.23 ± 1.0 and 4.60 ± 1.4 %ID/g at 30 min p.i.) as well as good retention in tumors (4.52 ± 0.3 and 3.13 ± 0.2 %ID/g at 2 h p.i.). [^68^Ga]**75** had a slow blood clearance despite its hydrophilicity, which led to the low value of the tumor/blood ratio at 2 h p.i. (0.49 ± 0.1). However, tumor/muscle ratios of [^68^Ga]**75** were still over one, and at 2 h p.i. [^68^Ga]**75** reached a tumor/muscle ratio of 3.11 ± 1.1. On the other hand, [^68^Ga]**76** exhibited both tumor/blood and tumor/muscle ratios (8.13 ± 1.7 and 1.45 ± 0.1 at 2 h p.i., respectively) over one due to low binding to serum proteins. Compared to [^18^F]FMISO (T/M = 3.85 ± 0.2, T/B = 4.84 ± 0.9) at 2 h p.i., [^68^Ga]**76** exhibited a higher tumor/blood ratio but a lower tumor/muscle ratio. 

The physical properties and biological evaluation of the ^68^Ga-labeled radiopharmaceuticals are summarized in Table 2. 

## 3. Conclusions and Perspectives

Hypoxia is commonly observed in tumors and impacts cancer treatment negatively. Therefore, accurate detection of tumor hypoxia is important in improving cancer treatments. Molecular imaging methods, particularly PET imaging, have been used for detecting tumor hypoxia. Recently, numerous PET radiotracers targeting hypoxia have been developed and studied in vitro and in vivo, providing valuable insights into this phenomenon. For example, [^18^F]FMISO has been commonly utilized in clinical studies to assess hypoxia levels. However, it has certain limitations due to its high lipophilicity. Therefore, there is a need for the development of novel PET radiotracers to improve physicochemical and biological properties and the effectiveness of hypoxia imaging.

The use of ^68^Ga-labeled radiopharmaceuticals has achieved significant success, with many of them utilizing derivatives of the ligand DOTA as bifunctional chelating agents to form octahedral coordination with gallium-68. Specifically, gallium-68 coordination usually involves four amines and two carboxylate groups, while the linkers connecting the chelating moieties and nitroimidazole moieties can be located on the free carboxylic acid groups or the cyclododecane rings. Many of the mentioned ^68^Ga-labeled radiopharmaceuticals bearing DOTA and NOTA derivatives have demonstrated good pharmacokinetic properties and selective targeting of tumor hypoxia. 

It appears that high hydrophilicity is a common characteristic of ^68^Ga-labeled radiotracers, primarily due to the presence of the hydrophilic groups on DOTA, NOTA, or TRAP derivatives. High hydrophilicity often enhances the clearance of the free radiotracer from blood, thereby improving the tumor/blood ratios and the contrast of PET images, particularly around the tumor sites. Moreover, rapid clearance of the radiotracer reduces the acquisition time required to obtain PET imaging results, thus minimizing patients’ exposure to radiation [125]. Radiotracers with high hydrophilicity are more likely to be absorbed and excreted via the kidneys rather than the liver. The high renal uptakes of ^68^Ga-labeled radiotracers can be maintained within safe levels and not affect PET imaging [126]. However, the highest hydrophilicity might not guarantee the superior pharmacokinetics of the radiotracer, as lipophilicity is also important to some extent. Adequate lipophilicity is necessary for the radiotracer to easily enter both normal and tumor cells via passive diffusion. In addition, if the radiotracer is cleared too rapidly, it might not have sufficient time to accumulate and retain in the tumor. 

With the use of ^68^Ge/^68^Ga generators, the preparation of ^68^Ga-labeled radiopharmaceuticals has become more convenient compared to other positron emitters like copper-64, fluorine-18, or carbon-11 which require a cyclotron. When compared to the commonly used radionuclide for PET, fluorine-18 has a half-life of 109.8 min, and gallium-68 has a shorter half-life of 68 min. However, both can be usefully used for PET studies as there is no great difference between their half-lives.

In several comparison studies, ^68^Ga-labeled radiopharmaceuticals are noninferior or comparable to the ^18^F-labeled radiopharmaceuticals, while in some studies the superior radiopharmaceutical depends on factors such as PET scanners and the physical and biological properties of the radiopharmaceutical [127,128,129].

In the future, the optimization of ^68^Ga-labeled radiopharmaceuticals’ structures can be approached from various directions like the length of the linker, the number of nitroimidazole moieties, and the position of the nitro group on imidazole heterocycles, as well as the presence of hydrophilic groups such as carboxylic acids and hydroxy groups in the structure. In addition, the utilization of novel bifunctional chelators in the optimization of ^68^Ga-labeled radiopharmaceuticals is also very promising. Each ^68^Ga-labeled radiopharmaceutical discussed in this review has its advantages and drawbacks in terms of lipophilicity, in vitro cellular uptake, in vivo tumor and normal tissue uptake and retention, and tumor/normal tissue contrasts. As a result, there is an ongoing need for the development of novel hypoxia-targeting PET radiotracers, particularly new hypoxia-targeting ^68^Ga-labeled radiopharmaceuticals with enhanced properties for future clinical use. This review provides a comprehensive overview of recent advancements in novel radiopharmaceuticals using gallium-68 for hypoxia imaging.

## Data Availability

Not applicable.

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
