# Peer review of "Recent Advances of 68Ga-Labeled PET Radiotracers with Nitroimidazole in the Diagnosis of Hypoxia Tumors"

_ijms, 2023, doi:10.3390/ijms241310552_

Round 1
Reviewer 1 Report
Authors present a historical review on the Ga-68 radiolabelling in order to synthesize radiopharmacdeuticals based on nitroimidazole (and not nitroimizazole) for PET imaging of hypoxia from 2009 to 2021.
Each article is resumed related to complex synthesis, hyodrophylic properties, as well as in vitro and in vivo biodistribution , tumor uptake, and pharmacokinetic properties of the developed molecules.
The introduction is consistent and cites lots of articles not related directly to hypoxia and the subject.
It would be very useful for the reader to understand which is the challenge or the need to develop Ga-68 labelled radiopharmaceuticals for PET hypoxia imaging, becouse F-18 MISO, F-18 FAZA and I-125 IAZA have been previously developped with lots of clinical trials and clinical utilites for hypoxia research well before 1999. I would appreciate a critical review of available data on F-18 nitroimidazole compounds currenly used for PET imaging a review of their advantages and disadvantages. The authors need to better justify the need and the potential utility of Ga-68 labelling versus F-18 labelging. Indeed Ga-68 nitroimidazole is more hydrophilic than F-18 nitroimidazole with more intense activity in kideny and bladder- wouldn't that be a radioprotection problem?
The presentation of the review is entitled Results. Title seem not appropriate. In fact, this is the review of the literature.From 97 references, only the last 12 are really in sibject with the review.
Each article and each radiopharmaceutical is commented in a chronological manner.
I would appreciate a Table synthesizing the main aspects related to each of the radiopharmaceuticals , in relation with stability, affinity, hydrophile properties, in vitro studies, in vivo studies, details on animal models and intensity of tumor uptake in hypoxic regions versus control regions, and whether studies performed in human. This comprehensiove synthetic table would ease lecture and help better apprehend the subject.
A table comparing the developped Ga based RP with F-18 FAZA and F-18 MISO for all presented compounds, (to highlight, when available), would also be necessary as this would bring arguments in favor of Ga-68 labeleld compounds. What about Cu-60 ATSM or F-18 FETNIM?
Finallly, there is only one clinical trial conducted in 20 patients with cervical cancer which needs to be more detailed as it is a clinical application in human subjects.The authors shoud detail on which of the radiopharmaceutical named on the previous paragraph was used in the clinical setting,detail the imaging procedure, the activity administered to patients, that images were obtained 30 and 60min after patient injection, etc. Comments related to possible explanations on the lack of correlation with hypoxia HIF-1 alpha expression would be interesting
The paper lacks arguments and a critical view of the subject. The Cahpter Discussion and perpsectives needs to be more devloped in a critical manner. Arguments in favor of Ga -68 versus others for hypoxia labelling are not evident and need to be detailed.
Ok the text is easy to read and comprehend
Author Response
Thank you for considering our revised manuscript entitled “68Ga-labeled PET Radiotracers with Nitroimizazole in the Diagnosis of Hypoxia Tumor: A Review (=>New title: Recent Advances in 68Ga-labeled PET Radiotracers with Nitroimidazole in the Diagnosis of Hypoxia Tumor)” for publication in International Journal of Molecular Sciences.
We would like to start by thanking the reviewers for their constructive criticisms and encouraging remarks.
You will find below that we answered all the reviewers’ questions and followed their suggestions by adding more descriptions and explanations.
(Q1) For the comment, “1. " The introduction is consistent and cites lots of articles not related directly to hypoxia and the subject.." ”,
(A1) The introduction contains explanations about molecular imaging, PET, hypoxia, nitroimidazole, 68Ga, Therefore, many references in introduction section are for molecular imaging, PET, hypoxia, nitroimidazole, 68Ga, not only hypoxia.
(Q2) For the comment, “2. "It would be very useful for the reader to understand which is the challenge or the need to develop Ga-68 labelled radiopharmaceuticals for PET hypoxia imaging, becouse F-18 MISO, F-18 FAZA and I-125 IAZA have been previously developped with lots of clinical trials and clinical utilites for hypoxia research well before 1999." ”,
(A2) Previously developed radiopharmaceuticals for PET hypoxia imaging were extensively studied but they have several drawbacks, and ideal radiopharmaceutical has not been found. Therefore, a critical review of developed radiopharmaceuticals was added to pages 2 and 3.
The following sentences were added into page 4 to provide understanding which is the challenge or the need to develop Ga-68 labelled radiopharmaceuticals for PET hypoxia imaging.
“…Generally, 18F-labeled radiotracers have been extensively studied due to the appropriate half-life time of 109.8 min and the ability to be incorporated into small and large molecules [109]. Moreover, 68Ga is produced via the use of 68Ge/68Ga generators. Therefore, preparation of 68Ga-labeled radiopharmaceuticals has become more convenient compared to other positron emitters like fluorine-18, carbon-11, or copper-64, which require a cyclotron. For example, despite the widespread use of 18F-labeled radiotracers in PET imaging, radiosyntheses of 18F-labeled radiotracers mostly rely on the production of fluorine-18 using cyclotron, which is big size and not widely available [110] …”
(Q3) For the comment, “3. " I would appreciate a critical review of available data on F-18 nitroimidazole compounds currenly used for PET imaging a review of their advantages and disadvantages." ”,
(A3) The following sentences were added into pages 2 and 3 to provide a critical review of available data on F-18 nitroimidazole compounds currently used for PET imaging a review of their advantages and disadvantages:
“…[18F]FMISO is the most common PET tracer for hypoxia imaging owing to its simple synthetic process and selectivity toward hypoxia. [18F]FMISO has been widely studied for hypoxia imaging of many types of cancers in both preclinical and clinical studies [79–84]. In the development of novel PET imaging agents, [18F]FMISO is still the standard for hypoxia and used as the control for hypoxia in many studies [85–87]. However, [18F]FMISO has some drawbacks such as slow blood clearance and high uptake values in the liver and gut due to its lipophilicity (logP value about 0.4) and slow tumor accumulation [50]. These drawbacks result in low-contrast PET images and extended optimal acquisition time, limiting its utility in clinical studies [88,89]. Thus, in order to improve the sensitivity and convenience of [18F]FMISO, the second generation of PET imaging agents for hypoxia was developed, namely [18F]FAZA and [18F]FETNIM. These radiotracers are also analogues of 2-nitroimidazole but are more hydrophilic than [18F]FMISO [90,91]. [18F]FAZA exhibited good uptakes and acceptable tumor/blood ratios in patients with lung, head and neck tumors, lymphoma, and gliomas [92]. Along with [18F]FAZA, [124I]IAZA is also another alternative to [18F]FMISO with better tumor/blood contrasts at early time post injection [93–96]. [18F]FETNIM was also tested clinically in patients with lung cancer, glioma, cervical carcinoma, esophageal and pancreatic cancer [97]. For example, the use of [18F]FETNIM PET/CT enabled the visualization of hypoxia in patients with esophageal squamous cell carcinoma. Recently, when tested in patients with head and neck cancer, [18F]FETNIM PET/CT was also successfully used for the selection of patients with hypoxia predicting poor overall survival [98]. However, when compared to [18F]FMISO in clinical studies in patients with lung cancer, [18F]FETNIM failed to show higher tumor/blood ratios than [18F]FMISO [99] (Table 1).
Additionally, [18F]EF5 was developed as fluorinated radiotracer more lipophilic than [18F]FMISO and the other second-generation radiotracers with logP values of 5.7 which enables faster distribution into normal and tumor tissues yet slower clearance from blood [100]. [18F]EF5 is a potential PET tracer with favorable tumor/muscle ratios when used for the imaging of hypoxia in patients with head and neck cancers. [18F]EF3, an analogue of [18F]EF5 with medium lipophilicity (logP = 1.25) was reported, however, when tested in mice and rat models, [18F]EF3 was not superior to [18F]FMISO in tumor/muscle ratios [101–103].
Later, [18F]HX4 and [18F]FRP170 were prepared as PET tracer for hypoxia is [18F]HX4 and [18F]FRP170. [18F]HX4 was clinically tested for patients with lung, head and neck, esophageal and pancreatic cancer, and showed comparable effective dose, improved contrast, and more favorable hydrophilicity (logP = –0.69) in comparison to [18F]FMISO, thus, frequent bladder voiding was required for patients due to the rapid renal clearance characteristic of [18F]HX4 [104,105]. [18F]FRP170 contains hydrophilic groups, thus, it was more hydrophilic than [18F]FMISO (logP = 0.094). It demonstrated good tumor uptake and tumor/background contrast in PET images as well as rapid clearance via kidneys when tested in humans with lung cancers [106,107].
Besides, Cu-ATSM, a lipophilic hypoxia radiotracer not containing nitroimidazole moieties, was developed. It can be radiolabeled with several radionuclides like 64Cu, 60Cu, and 62Cu. When tested in patients with cervical carcinoma, 64Cu-ATSM showed better-quality PET images with lower noise compared to 60Cu-ATSM [108]. Furthermore, numerous novel radiotracers have been developed to enhance the efficacy of current radiopharmaceuticals, and numerous preclinical studies have been conducted….”
(Q4) For the comment, “3. " The authors need to better justify the need and the potential utility of Ga-68 labelling versus F-18 labelging. Indeed Ga-68 nitroimidazole is more hydrophilic than F-18 nitroimidazole with more intense activity in kideny and bladder- wouldn't that be a radioprotection problem?" ”,
(A4) The following sentences were added into page 4 to provide the need and the potential utility of Ga-68 labeling versus F-18 labeling (it is similar comments for #2):
“…Generally, 18F-labeled radiotracers have been extensively studied due to the appropriate half-life time of 109.8 min and the ability to be incorporated into small and large molecules [109]. Moreover, 68Ga is produced via the use of 68Ge/68Ga generators. Therefore, preparation of 68Ga-labeled radiopharmaceuticals has become more convenient compared to other positron emitters like fluorine-18, carbon-11, or copper-64, which require a cyclotron. For example, despite the widespread use of 18F-labeled radiotracers in PET imaging, radiosyntheses of 18F-labeled radiotracers mostly rely on the production of fluorine-18 using cyclotron, which is big size and not widely available [110]
Developing more hydrophilic radiotracers compared to [18F]FMISO is currently the primary need in the development of novel hypoxia-targeting radiopharmaceuticals. 68Ga, a radionuclide with a half-life of 68 minutes, has also been applied for PET imaging. Bifunctional chelates have been widely employed to generate octahedral coordination with the gallium-68 radionuclide at the same time to bear the nitroimidazole moieties. These chelates often contain hydrophilic moieties which contribute to increasing the hydrophilicity of the 68Ga-labeled complex….”
In addition, the following sentences were added into page 17 (Conclusion and perspective) to discuss on the hydrophilicity of Ga-68 nitroimidazole and F-18 nitroimidazole and the effect on the kidney and bladder.
“…It appears that high hydrophilicity is a common characteristic of 68Ga-labeled radiotracers, primarily due to the presence of the hydrophilic groups on DOTA, NOTA, or TRAP derivatives. High hydrophilicity often enhances the clearance of the free radiotracer from blood, thereby improving the tumor/blood ratios and the contrast of PET images, particularly around the tumor sites. Moreover, rapid clearance of the radiotracer reduces the acquisition time required to obtain PET imaging results, thus minimizing patients’ exposure to radiation [125]. Radiotracers with high hydrophilicity are more likely to be absorbed and excreted via the kidneys rather than the liver. The high renal uptakes of 68Ga-labeled radiotracers can be maintained within safe levels and not affect PET imaging [126].”
In addition, in clinical studies of a renal excretion-promoting radiotracer, [18F]HX4, patients were provided adequate hydration and frequent bladder voiding. The following sentences were added into page 3:
“…[18F]HX4 was clinically tested for patients with lung, head and neck, esophageal and pancreatic cancer, and showed comparable effective dose, improved contrast, and more favorable hydrophilicity (logP = –0.69) in comparison to [18F]FMISO, thus, frequent bladder voiding was required for patients due to the rapid renal clearance characteristic of [18F]HX4 [104,105]….”
(Q5) For the comment, “4. " The presentation of the review is entitled Results. Title seem not appropriate. In fact, this is the review of the literature. From 97 references, only the last 12 are really in sibject with the review.." ”,
(A5) The Title “68Ga-labeled PET Radiotracers with Nitroimizazole in the Diagnosis of Hypoxia Tumor: A Review” were changed to “Recent Advances of 68Ga-labeled PET Radiotracers with Nitroimizazole in the Diagnosis of Hypoxia Tumor”.
(Q6) For the comment, “5. " I would appreciate a Table synthesizing the main aspects related to each of the radiopharmaceuticals, in relation with stability, affinity, hydrophile properties, in vitro studies, in vivo studies, details on animal models and intensity of tumor uptake in hypoxic regions versus control regions, and whether studies performed in human. This comprehensiove synthetic table would ease lecture and help better apprehend the subject..." ”,
(A6) Table 2 synthesizing the main aspects was inserted into page 14.
(Q7) For the comment, “6. " A table comparing the developped Ga based RP with F-18 FAZA and F-18 MISO for all presented compounds, (to highlight, when available), would also be necessary as this would bring arguments in favor of Ga-68 labeleld compounds. What about Cu-60 ATSM or F-18 FETNIM?..." ”,
(A7) Table 2 already contains the comparison of the developed Ga-based RP with F-18 FAZA and F-18 MISO and it was inserted into page 14.
Besides, all of articles containing the developed Ga based RP did not mention about comparison of Ga-68 labeled compounds with Cu-60 ATSM or F-18 FETNIM
(Q8) For the comment, “7. "Finallly, there is only one clinical trial conducted in 20 patients with cervical cancer which needs to be more detailed as it is a clinical application in human subjects. The authors shoud detail on which of the radiopharmaceutical named on the previous paragraph was used in the clinical setting, detail the imaging procedure, the activity administered to patients, that images were obtained 30 and 60min after patient injection, etc.." ”,
(A8) The name of radiopharmaceutical was detailed as [68Ga]31.
The following sentences were added into page 9-10 to give more details on the clinical study.
“…The patients were injected with 68Ga-31, with injected activity ranging from 1.8 to 2.2 MBq/kg. Pelvic images were taken at 30 min p.i. and the PET/CT images were taken at 60 min p.i. after urinary catheterization. 18F-FDG PET/CT studies were performed on the same 20 patients on separate days to 68Ga-nitroimidazole PET/CT, with the injected activity of 3–5 MBq/kg….”
(Q9) For the comment, 8. " Comments related to possible explanations on the lack of correlation with hypoxia HIF-1 alpha expression would be interesting..." ”,
(A9) The following sentences were also added into page 10 to give more information of the unexpected results.
“….which might be explained by the heterogeneity of hypoxia, resulting in a smaller volume of hypoxia than the tumor volume. Notably, hypoxic tumor volume was negatively correlated with immunohistochemistry, which might be due to the feedback mechanism that occurred between HIF-1α and markers or genes of the tumor, or the presence of the other isoforms of HIF-1α, such as HIF-2α and HIF-3α…”
(Q10) For the comment, “9. "The paper lacks arguments and a critical view of the subject. The Cahpter Discussion and perpsectives needs to be more devloped in a critical manner. Arguments in favor of Ga -68 versus others for hypoxia labelling are not evident and need to be detailed." ”,
(A10) The following sentences were added into the Conclusion and Perspectives section in order to present the arguments and critical view of the hydrophilicity of the 68Ga-labeled radiotracers, as well as comparing gallium-68 with the common PET radionuclide fluorine-18:
“….It appears that high hydrophilicity is a common characteristic of 68Ga-labeled radiotracers, primarily due to the presence of the hydrophilic groups on DOTA, NOTA, or TRAP derivatives. High hydrophilicity often enhances the clearance of the free radiotracer from blood, thereby improving the tumor/blood ratios and the contrast of PET images, particularly around the tumor sites. Moreover, rapid clearance of the radiotracer reduces the acquisition time required to obtain the PET imaging results, thus minimizing patients’ exposure to radiation [125]. Radiotracers with high hydrophilicity are more likely to be absorbed and excreted via the kidneys rather than the liver. The high renal uptakes of 68Ga-labeled radiotracers can be maintained within safe levels and not affect PET imaging [126]. However, the highest hydrophilicity might not guarantee superior pharmacokinetics of the radiotracer, as lipophilicity is also important to some extent. Adequate lipophilicity is necessary for the radiotracer to easily enter both normal and tumor cells via passive diffusion. In addition, if the radiotracer is cleared too rapidly, it might not have sufficient time to accumulate and retain in the tumor.
With the use of 68Ge/68Ga generators, the preparation of 68Ga-labeled radiopharmaceuticals has become more convenient compared to other positron emitters like copper-64, fluorine-18, or carbon-11 which require a cyclotron. When compared to the commonly used radionuclide for PET, fluorine-18 has a half-life of 110 minutes, and gallium-68 has a shorter half-life of 68 minutes. However, both can be usefully used for PET studies due to not big different half-life.
In several comparison studies, 68Ga-labeled radiopharmaceuticals are non-inferior or comparable to the 18F-labeled radiopharmaceuticals, while in some studies the superior radiopharmaceutical depends on factors such as PET scanners, physical and biological properties of the radiopharmaceutical [127–129].
In the future, optimization of 68Ga-labeled radiopharmaceuticals’ structures can be approached from various directions like the length of the linker, the number of nitroimidazole moieties, the position of the nitro group on imidazole heterocycles, as well as the presence of hydrophilic groups such as carboxylic acids and hydroxy groups in the structure. In addition, the utilization of novel bifunctional chelators in the optimization of 68Ga-labeled radiopharmaceuticals is also very promising….”
We hope that our modifications to the manuscript for the specific concerns and questionable points will satisfy the reviewers and the requirements for the publication of this manuscript.

Reviewer 2 Report
Journal of International Journal of Molecular Sciences
Review Article;
The article entitled “68Ga-labeled PET Radiotracers with Nitroimizazole in the Diagnosis of Hypoxia Tumor: A Review’’. The author reviews Positron emission tomography. Which is a non-invasive molecular imaging method which is extensively applied in the detection and treatment of various diseases. As hypoxia is a common phenomenon found in most of solid tumors. Nitroimidazole is a group of bioreducible pharmacophores that selectively accumulate in hypoxic regions of the body. As Gallium-68, a positron-emitting radioisotope, has a favorable half-life time of 68 minutes and can be produced conveniently by 68Ge/68Ga generators. Recently, there is significant progress in the preparation of novel 68Ga-labeled complexes bearing nitroimidazole moieties for the diagnosis of hypoxia. This study provides a comprehensive overview of the current status of developing 68Ga-labeled radiopharmaceuticals with nitroimidazole moieties.
I carefully read the manuscript and I accept it for possible publication in IJMS. There are some common mistakes, references and English language problems in the article which should be corrected by the authors. After correcting the mistakes, the article could be considered for publication in the prestigious International Journal of Molecular Sciences Journal.
Comments for Authors
Ø The author needs to revise the title of the manuscript.
Ø Write keywords in alphabetical order.
Ø The author needs to include the previously used biomarker used for hypoxia. Include one table of approved and under-trial hypoxia biomarkers.
Ø The author needs to include the mechanism of action of Gallium-68in a graphical image.
Ø There are some grammatical mistakes, the author needs to revise the manuscript.
Cite the following references;
v DOI: 10.2174/1871520622666220831124321
v https://doi.org/10.1038/s41419-021-03771-z

Author Response
Thank you for considering our revised manuscript entitled “68Ga-labeled PET Radiotracers with Nitroimizazole in the Diagnosis of Hypoxia Tumor: A Review (=>New title: Recent Advances in 68Ga-labeled PET Radiotracers with Nitroimidazole in the Diagnosis of Hypoxia Tumor)” for publication in International Journal of Molecular Sciences.
We would like to start by thanking the reviewers for their constructive criticisms and encouraging remarks.
You will find below that we answered all the reviewers’ questions and followed their suggestions by adding more descriptions and explanations.
(Q1) For the comment, “1. " The author needs to revise the title of the manuscript...." ”,
(A1) The title “68Ga-labeled PET Radiotracers with Nitroimizazole in the Diagnosis of Hypoxia Tumor: A Review” were changed to “Recent Advances of 68Ga-labeled PET Radiotracers with Nitroimidazole in the Diagnosis of Hypoxia Tumor”.
(Q2) For the comment, “2. " Write keywords in alphabetical order.." ”,
(A2) Keywords were put in alphabetical order.
(Q3) For the comment, “3. " The author needs to include the previously used biomarker used for hypoxia. Include one table of approved and under-trial hypoxia biomarkers...." ”,
(A3) The following sentences were added into pages 2 and 3 to provide the previously used biomarker used for hypoxia, and Table 1 was inserted into page 3.
“…[18F]FMISO is the most common PET tracer for hypoxia imaging owing to its simple synthetic process and selectivity toward hypoxia. [18F]FMISO has been widely studied for hypoxia imaging of many types of cancers in both preclinical and clinical studies [79–84]. In the development of novel PET imaging agents, [18F]FMISO is still the standard for hypoxia and used as the control for hypoxia in many studies [85–87]. However, [18F]FMISO has some drawbacks such as slow blood clearance and high uptake values in the liver and gut due to its lipophilicity (logP value about 0.4) and slow tumor accumulation [50]. These drawbacks result in low-contrast PET images and extended optimal acquisition time, limiting its utility in clinical studies [88,89]. Thus, in order to improve the sensitivity and convenience of [18F]FMISO, the second generation of PET imaging agents for hypoxia was developed, namely [18F]FAZA and [18F]FETNIM. These radiotracers are also analogues of 2-nitroimidazole but are more hydrophilic than [18F]FMISO [90,91]. [18F]FAZA exhibited good uptakes and acceptable tumor/blood ratios in patients with lung, head and neck tumors, lymphoma, and gliomas [92]. Along with [18F]FAZA, [124I]IAZA is also another alternative to [18F]FMISO with better tumor/blood contrasts at early time post injection [93–96]. [18F]FETNIM was also tested clinically in patients with lung cancer, glioma, cervical carcinoma, esophageal and pancreatic cancer [97]. For example, the use of [18F]FETNIM PET/CT enabled the visualization of hypoxia in patients with esophageal squamous cell carcinoma. Recently, when tested in patients with head and neck cancer, [18F]FETNIM PET/CT was also successfully used for the selection of patients with hypoxia predicting poor overall survival [98]. However, when compared to [18F]FMISO in clinical studies in patients with lung cancer, [18F]FETNIM failed to show higher tumor/blood ratios than [18F]FMISO [99] (Table 1).
Additionally, [18F]EF5 was developed as fluorinated radiotracer more lipophilic than [18F]FMISO and the other second-generation radiotracers with logP values of 5.7 which enables faster distribution into normal and tumor tissues yet slower clearance from blood [100]. [18F]EF5 is a potential PET tracer with favorable tumor/muscle ratios when used for the imaging of hypoxia in patients with head and neck cancers. [18F]EF3, an analogue of [18F]EF5 with medium lipophilicity (logP = 1.25) was reported, however, when tested in mice and rat models, [18F]EF3 was not superior to [18F]FMISO in tumor/muscle ratios [101–103].
Later, [18F]HX4 and [18F]FRP170 were prepared as PET tracer for hypoxia is [18F]HX4 and [18F]FRP170. [18F]HX4 was clinically tested for patients with lung, head and neck, esophageal and pancreatic cancer, and showed comparable effective dose, improved contrast, and more favorable hydrophilicity (logP = –0.69) in comparison to [18F]FMISO, thus, frequent bladder voiding was required for patients due to the rapid renal clearance characteristic of [18F]HX4 [104,105]. [18F]FRP170 contains hydrophilic groups, thus, it was more hydrophilic than [18F]FMISO (logP = 0.094). It demonstrated good tumor uptake and tumor/background contrast in PET images as well as rapid clearance via kidneys when tested in humans with lung cancers [106,107].
Besides, Cu-ATSM, a lipophilic hypoxia radiotracer not containing nitroimidazole moieties, was developed. It can be radiolabeled with several radionuclides like 64Cu, 60Cu, and 62Cu. When tested in patients with cervical carcinoma, 64Cu-ATSM showed better-quality PET images with lower noise compared to 60Cu-ATSM [108]. Furthermore, numerous novel radiotracers have been developed to enhance the efficacy of current radiopharmaceuticals, and numerous preclinical studies have been conducted….”
(Q4) For the comment, “4. " The author needs to include the mechanism of action of Gallium-68 in a graphical image”,
(A4) The mechanism of action of Gallium-68 was drawn as a graphical image (Figure 1) and inserted into page 4.
(Q5) For the comment, “5. " There are some grammatical mistakes, the author needs to revise the manuscript...." ”,
(A5) We checked and corrected some grammatical mistakes, and revised the manuscript.
(Q6) For the comment, “6. " Cite the following references; DOI:10.2174/1871520622666220831124321, https://doi.org/10.1038/s41419-021-03771-z " ”,
(A6) The references (DOI:10.2174/1871520622666220831124321, https://doi.org/10.1038/s41419-021-03771-z) were inserted into page 2 (references number 60 and 61).
We hope that our modifications to the manuscript for the specific concerns and questionable points will satisfy the reviewers and the requirements for the publication of this manuscript.

Round 2
Reviewer 1 Report
I have read the new revised version.
Authors have well responded to all suggestions
It would be advisable to add references on the column with the clinical trials for each clinical trial ( Table)
Some minor grammar errors remain to be corrected.
Author Response
Thank you for considering our revised manuscript entitled “Recent Advances of 68Ga-labeled PET Radiotracers with Nitroimidazole in the Diagnosis of Hypoxia Tumor” for publication in International Journal of Molecular Sciences.
We would like to start by thanking the reviewers for their constructive criticisms and encouraging remarks.
You will find below that we answered all the reviewers’ questions and followed their suggestions by adding more descriptions and explanations.
(Q1) For the comment, “1. "It would be advisable to add references on the column with the clinical trials for each clinical trial (Table)" ”,
(A1) The reference [120] was added to the column of Table 2 in page 15, and the sentence for the radiopharmaceutical “68Ga-labeled trivalent complexes with TRAP chelator ([68Ga]28-31) become “Clinical application: Patients with cervical cancer No significant correlation of PET imaging results with the immunohistochemical HIF-1α positive regions [120]”
(Q2) For the comment, “2. " Some minor grammar errors remain to be corrected." ”,
(A2) Grammar errors were checked and corrected.
We hope that our modifications to the manuscript for the specific concerns and questionable points will satisfy the reviewers and the requirements for the publication of this manuscript.
